# Effect of a scaled-up quality improvement intervention on health workers' competence on neonatal resuscitation in simulated settings in public hospitals: A pre-post study in Nepal

**Dipak Raj Chaulagain**[1]*, **Ashish K. C.**[1,2], **Johan Wrammert**[1], **Olivia Brunell**[1], **Omkar Basnet**[3], **Mats Malqvist**[1]

1 Department of Women's and Children's Health, Uppsala Global Health Research on Implementation and Sustainability (UGHRIS), Uppsala University, Uppsala, Sweden, 2 Society of Public Health Physicians Nepal (SOPHPHYN), Kathmandu, Nepal, 3 Golden Community, Lalitpur, Nepal

* dipak.chaulagain@kbh.uu.se

## Abstract

### Background

Helping Babies Breathe (HBB) training improves bag and mask ventilation and reduces neonatal mortality and fresh stillbirths. Quality improvement (QI) interventions can improve retention of neonatal resuscitation knowledge and skills. This study aimed to evaluate the effect of a scaled-up QI intervention package on uptake and retention of neonatal resuscitation knowledge and skills in simulated settings.

### Methods

This was a pre-post study in 12 public hospitals of Nepal. Knowledge and skills of trainees on neonatal resuscitation were evaluated against the set standard before and after the introduction of QI interventions.

### Results

Altogether 380 participants were included for knowledge evaluation and 286 for skill evaluation. The overall knowledge test score increased from 14.12 (pre-basic) to 15.91 (post-basic) during basic training *(p < 0.001)*. The knowledge score decreased over time; 15.91 (post-basic) vs. 15.33 (pre-refresher) *(p < 0.001)*. Overall skill score during basic training (16.98 ± 1.79) deteriorated over time to 16.44 ± 1.99 during refresher training *(p < 0.001)*. The proportion of trainees passing the knowledge test increased to 91.1% (post-basic) from 67.9% (pre-basic) which decreased to 86.6% during refresher training after six months. The knowledge and skill scores were maintained above the set standard (>14.0) over time at all hospitals during refresher training.

**Data Availability Statement:** All relevant data are within the paper and its Supporting information files.

**Funding:** Funding was obtained from the Vetenskapsrådet (SE), the Laerdal Foundation for Acute Medicine, Norway, and Einhorn Family Foundation, Sweden. AKC was the grantee of Vetenskapsrådet (SE). MM was the grantee of Laerdal Foundation for Acute Medicine, Norway, and Einhorn Family Foundation, Sweden. Golden Community is not a commercial company but a national research agency and is the sub-grantee of the Swedish Research Council fund for data collection of Nepal Perinatal Quality Improvement Project (NePeriQIP). The funder provided support in form of salaries for OB, but did not have any additional role in the study design, data collection and analysis.

**Competing interests:** The authors have declared that no competing interest exist.

## Conclusion

HBB training together with QI tools improves health workers' knowledge and skills on neonatal resuscitation, irrespective of size and type of hospitals. The knowledge and skills deteriorate over time but do not fall below the standard. The HBB training together with QI interventions can be scaled up in other public hospitals.

## Trial registration

This study was part of the larger Nepal Perinatal Quality Improvement Project (NePeriQIP) with International Standard Randomised Controlled Trial Number, ISRCTN30829654, registered 17th of May, 2017.

## Introduction

Intrapartum hypoxic events, previously known as "birth asphyxia", continue to be a global challenge with an annual estimated 660,000 neonatal deaths on its account [1]. It may lead to long-term consequences as cerebral palsy, epilepsy, and learning disabilities in surviving infants [2, 3]. Most of these deaths would have been averted with early-initiated simple and low-cost resuscitation support [4, 5]. Therefore, neonatal resuscitation competency among health workers remains critical in saving neonates who cannot initiate breathing at birth [6]. Helping Babies Breathe (HBB) is a simulation-based neonatal resuscitation program developed by the American Academy of Pediatrics especially for health workers in resource-limited settings [7, 8]. HBB has demonstrated a significant positive impact on early neonatal mortality including fresh stillbirth and first-day neonatal mortality [9–11]. The HBB program has demonstrated improvement in bag and mask ventilation in low-and middle-income countries [12]. However, HBB training alone, without combining it with ongoing training and continuous mentoring, is not sufficient to transfer health workers' simulated skills into clinical practice [9, 13]. Several pre-post studies have demonstrated improvement in trainees' skills after HBB training but a significant decrease in knowledge and skills over time has also been reported [2, 14]. Therefore, sustainable uptake, retention, and application of knowledge and skills on neonatal resuscitation and its impact on newborn survival is still a major question in resource-limited settings.

Quality improvement (QI) interventions including frequent skill practice, ongoing training, monitoring, and professional support are indicated to increase the likelihood of sustaining neonatal resuscitation knowledge and skills and advancing gains in neonatal outcomes [15–19]. Advanced gains in neonatal outcomes post-HBB training were found to be associated with QI interventions in Kenya [1]. The quality of care intervention in Tanzania improved the overall index scores for the quality of observed essential newborn care [20]. A recent small-scale study in Peru reported the retained and even improved HBB knowledge and skills with simple, inexpensive interventions, including supervised training [17]. Supportive supervision and regular follow-up visits were considered important aspects for the retention of neonatal resuscitation skills in Tanzania [21]. Similarly, an in-depth refresher course provided to midwives slowed the deterioration of neonatal resuscitation skills in rural Ghana [22].

The Government of Nepal has pledged to improve the quality of neonatal care services at the point of delivery to achieve the target of reducing neonatal deaths [23]. More effective and efficient implementation approaches need to be identified for ensuring the quality of neonatal

resuscitation services [8]. The HBB linked with a QI package decreased intrapartum stillbirth and first-day neonatal mortality in a referral maternity hospital in Nepal [16]. Health workers practicing bag and mask skills, preparing for resuscitation before every birth, using self-evaluation checklists, and attending weekly review meetings were more likely to retain neonatal resuscitation skills in the same hospital [15]. These findings indicated the need of scaling up and testing the adaptability of the QI package in other health facilities in the existing health system to generate more insight on uptake and retention of neonatal resuscitation knowledge and skills [15].

Building on the success of the HBB Quality improvement intervention package in a referral maternity hospital in Nepal, the Ministry of Health and Population together with the study team developed a QI intervention package. We scaled up this package in 12 secondary-level public hospitals in Nepal. This study aimed to evaluate the effect of this scaled-up QI intervention on uptake and retention of neonatal resuscitation knowledge and skills among health workers in simulated settings.

## Materials and methods

### Study design

This study was part of the larger Nepal Perinatal Quality Improvement Project (NePeriQIP) [15, 24]. Nested within the NePeriQIP (ISRCTN30829654), we used a pre-post study design to evaluate the effect of a scaled-up quality improvement intervention on resuscitation knowledge and skills among health workers. The study was conducted from July 2017 to December 2018.

### Study settings

We conducted this study in 12 public hospitals in Nepal. The participating hospitals were categorized into three different sizes based on the volume of annual deliveries; i) high-volume hospitals, ii) medium-volume hospitals and iii) low-volume hospitals. The hospitals with > 8000 deliveries per year were grouped into high-volume, hospitals with 3000 to 7999 deliveries a year were grouped into medium-volume, and hospitals with <3000 deliveries a year were grouped into low-volume hospitals. There were four hospitals in each category.

### QI intervention

The QI intervention was initiated at different time points in hospitals according to the stepped wedge study design of NePeriQIP [24]. Each wedge consisted of three hospitals (one high-volume, one medium-volume, and one low-volume hospital). In collaboration with the Ministry of Health and the respective hospitals, we introduced the QI interventions (S1 Text) in the following stages in all hospitals;

1. **Orientation** to the hospital management team on the QI package on perinatal care.

2. **Selection of in-hospital QI facilitators and external mentors;** The hospital management committee selected the in-hospital QI facilitators from among the pediatricians, medical officers, and nursing staff. The number of facilitators depended on the size of the hospital; two from low-volume, three from medium-volume, and four from high-volume-hospitals. The study team recruited external mentors from among senior pediatricians, medical officers, and matrons to provide continuous support to the hospital team during the implementation of QI interventions.

3. **Master training of trainers;** A seven-day master training was organized per wedge for QI facilitators. The in-hospital QI facilitators were trained with the package of QI intervention

including HBB. Together with the in-hospital QI facilitators, the external mentors were also included in the training.

4. **Assessment of perinatal care services in the hospital;** The QI facilitators assessed the readiness and availability of perinatal care services using a pre-developed checklist. A two-day bottleneck analysis workshop was conducted at each hospital for neonatal services.

5. **On-site basic training to health workers;** The in-hospital QI facilitators together with external mentors conducted basic training in respective hospitals that was a three-day package. All health workers involved in perinatal care were included in this training. Health workers were allocated in different batches (20 participants per batch). Out of three days, the first day was fully dedicated to HBB. The HBB training manual version one translated into Nepali was used during the training. In compliance with the HBB training manual, we developed a training registration and course evaluation form to be used by the facilitators and the participants (S2 Text). The basic HBB training was supplemented by content on quality improvement intervention.

6. **Provision of QI tools;** Following the basic training, the hospitals were provided with the HBB job aid, self-assessment checklists, HBB mannequin set for the skill check, scoreboards, and weekly PDSA review meeting notes.

7. **Post-training QI practice;** After the basic training, the QI facilitators initiated weekly PDSA meetings in each hospital. Together with this, the health workers started daily skill checks on the bag and mask ventilation using the mannequin. Also, a scoreboard with major indicators on neonatal resuscitation was installed in the delivery ward which was updated daily.

8. **Refresher training;** Around six months after the basic training, health workers were enrolled in one-day refresher training on QI interventions with a special focus on HBB. A shortened version of the training registration and course evaluation form was used during this training.

## Participants

All health workers involved in perinatal care during the study period were included in the study. There was no a priori estimation of the sample size because the participants were selected by the respective hospital.

## Variables

The outcome variables under this study were; a) knowledge score and b) skill score obtained by trainees for neonatal resuscitation.

## Data collection procedures

Data related to participants' knowledge on neonatal resuscitation were collected at three points; i) before starting the basic training (pre-basic); ii) at the end of basic training (post-basic); iii) after six months during refresher training (pre-refresher). The standard set of 17 multiple-choice questions were administered to assess the knowledge during basic and refresher training. The participants took the objective structured clinical examination B (OSCE B) with a mannequin during basic and refresher training. OSCE B presented a late pre-term infant born in secondary apnoea who required positive pressure ventilation to survive.

OSCE B comprised 18 skill-related questions. Each trainee recorded responses to the knowledge test and OSCE B test in an individual training registration and evaluation form.

## Data management

We recorded study data in paper forms (training registration and evaluation forms) at each hospital. The in-hospital QI facilitators and the external mentors did the consistency checks. The data management officer at the central research office in Kathmandu entered the data into the electronic database SPSS. The data collection forms have been securely stored after entering into the database.

## Evaluation of knowledge and skills

The correct response to 14 out of 17 questions ($\geq 82.35\%$) was set as the standard to pass the knowledge test. The mean knowledge test score and standard deviation were analyzed for hospitals by size, for individual hospitals, and the overall group. The differences in knowledge scores between the hospitals by size and for individual hospitals were also analyzed. The proportion of trainees passing the knowledge test was analyzed for hospitals by size and the overall group.

The participants performed the OSCE B test after the demonstration of its complete steps by the facilitators during basic training, but it was performed without demonstration during refresher training. To pass the skill test, the participants required correct responses to 14 out of 18 questions ($\geq 77.7\%$). The OSCE B scenario required four essential stages; i) recognizing that the newborn infant is not breathing, ii) ventilation at a rate of 40 breaths/minute, iii) looking for chest movement, and iv) improving ventilation if needed. The mean skill score and standard deviation were analyzed for hospitals by size, for individual hospitals, and the overall group. Similarly, the proportion of participants passing the skill test were analyzed for hospitals by size and the overall group.

## Statistical analyses

Descriptive statistics were calculated for background characteristics of the trainees by the size of the hospitals. We performed the paired comparison of the numeric scores obtained on the knowledge tests for the overall group, hospitals by size, and individual hospitals. Wilcoxon signed-rank test was performed for paired comparison between; i) pre-basic and post-basic, ii) post-basic and pre-refresher, and iii) pre-basic and pre-refresher test. Wilcoxon signed-rank test was performed for paired comparison of the numeric scores obtained on OSCE B during the basic training (post-basic) and refresher training (pre-refresher) for hospitals by size and for individual hospitals. Kruskal Wallis test was performed for comparison between hospitals by size and between individual hospitals. Mc Nemar test of proportions was used to test the paired differences in pass rates on the knowledge questionnaire and OSCE B according to the size of hospitals and for the overall group. A logistic regression model was used to assess the association between health workers' characteristics and two dependent variables; i) change in neonatal resuscitation knowledge level over time, and; ii) change in neonatal resuscitation skill level over time. The trainees who failed both the post-basic and pre-refresher tests or passed the post-basic test but failed pre-refresher test were categorized as having 'deteriorated or unimproved' knowledge or skill level. The trainees who passed both the post-basic and pre-refresher tests or failed the post-basic test but passed the pre-refresher test were categorized as having 'improved or retained' knowledge or skill level. The characteristics of the trainees used in the model were; the size of the associated hospitals, current position, age, gender, years of working experience, previous HBB training, previous experience of resuscitating newborn

infants, and experience of attending at least one delivery in the past. A p-value *of* less than 0.05 was considered statistically significant.

### Ethical considerations

The ethics approval was obtained from the Nepal Health Research Council (ref 26–2017). Verbal informed consent was obtained from all participants before starting basic and refresher training. All participants were provided with verbal information on their participation in the study. The training registration and evaluation form (S2 Text), which was used to evaluate trainees' knowledge and skills in this study, was used for the documentation of their consent. The participants filled and signed the 'training registration and evaluation form ' after agreeing to participate in the study. The process of obtaining consent was completed in the presence of hospital management in the respective hospitals. The signed individual training registration and evaluation forms have been securely stored at the central research office in Kathmandu.

### Results

Altogether 798 trainees participated in the basic training and 702 were in the refresher training (Fig 1). Out of the total, only 380 trainees were included in the knowledge evaluation and 286 were included for skill evaluation. The majority of the participants were nurses (60.9%) followed by Auxiliary nurse midwives (26.3%) (Table 1). Only 48.1% of the trainees had received neonatal resuscitation training before and 59.2% had an experience of resuscitating newborns in the past.

The baseline performance in the knowledge test (pre-basic test mean score) did not vary significantly between the hospitals by size *(p = 0.909)* but the score varied between the individual hospitals *(p = 0.001)* with five hospitals below the cut-off (<14.0). The overall average knowledge test score increased from 14.12 (pre-basic) to 15.91 (post-basic) during basic training *(p< 0.001)*. Improvement in knowledge score was observed in all size of hospitals; high-volume *(p < 0.001)*, medium-volume *(p < 0.001)* and low-volume *(p = 0.001)*. Similarly, there was an improvement in knowledge scores in individual hospitals except hospital 1 (Table 2). There was no difference in the post-basic test score between hospitals by size *(p = 0.568)*, but the difference was found between individual hospitals *(p <0.001)*. However, the post-basic test knowledge score was above the standard (>14.00) for all hospitals.

The overall knowledge score decreased over time; 15.91 (post-basic) vs. 15.33 (pre-refresher) *(p < 0.001)*. There was a decrease in knowledge for high-volume hospitals *(p = 0.036)*, medium-volume hospitals *(p = 0.002)* and low-volume hospitals *(p = 0.014)*. At the individual hospital level, the reduction was found only in four hospitals (Table 2). The knowledge score at refresher training did not differ between high, medium, and low-volume hospitals *(p = 0.061)*, but the difference was found between individual hospitals *(p < 0.001)*. However, the pre-refresher knowledge test score was maintained above the standard at all hospitals. Importantly, the overall average knowledge score was remarkably higher during refresher training compared to the pre-basic test score *(p < 0.001)*.

Overall skill score during basic training (16.98 ± SD 1.79) deteriorated over time to 16.44 ± 1.99 during refresher training *(p < 0.001)*. The deterioration of skill was observed in high-volume hospitals (p < 0.001). The average skill score was however above the cut-off value (>14.00) for all hospitals by size and for individual hospitals (Table 2).

Overall, there was an increase in the proportion of trainees who passed post-basic test (91.1%) compared to those who passed pre-basic (67.9%) *(p = 0.003)* (Table 3). The proportion of trainees who passed the post-basic test was higher than those who passed the pre-basic test

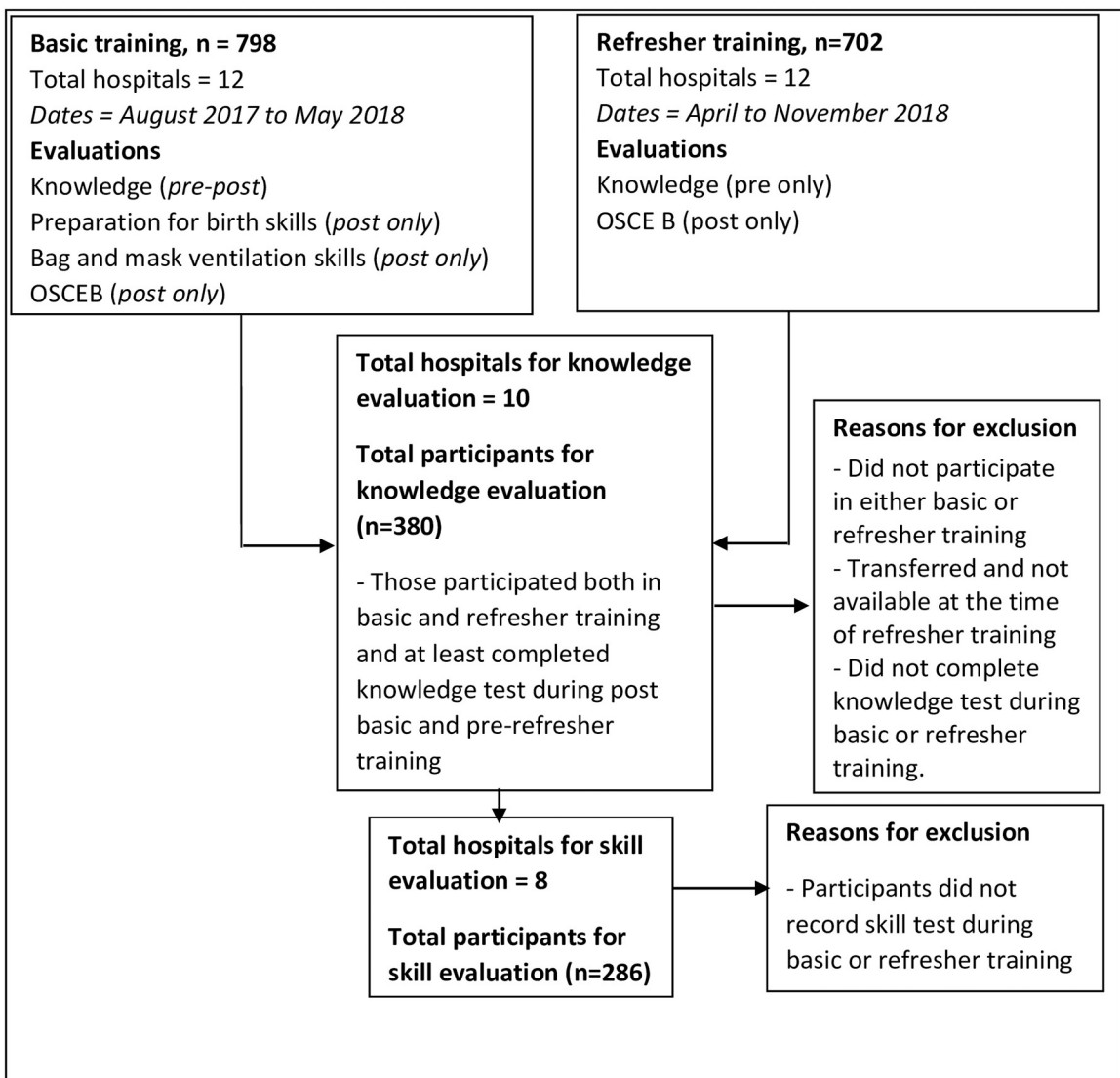

**Fig 1. Flow diagram of training participants.**

in all levels of hospitals. The overall passing rate decreased over time and was 86.6% in pre-refresher training (*p = 0.023*). Overall, the proportion of the trainees who passed skill tests also deteriorated over time; 94.4% post-basic vs. 87.8% pre-refresher test (*p = 0.002*). However, the difference was observed only for high-volume hospitals (*p = 0.001*) (Table 3).

The univariate logistic regression analysis showed that being male and having no experience of resuscitating newborns in the past were two factors associated with deteriorated or unimproved knowledge levels over time (Table 4). However, the association of these characteristics with the deterioration of knowledge was not confirmed when adjusted for other characteristics. Similarly, the deterioration of skills was found to be associated with previous experience of attending deliveries in univariate logistic regression; however, it was not confirmed when adjusted for other characteristics (Table 5).

**Table 1. Background characteristics of the training participants by the size of hospitals.**

| Characteristics | High volume hospitals (N = 252) | Medium volume hospitals (N = 97) | Low volume hospitals (N = 31) | Total |
|---|---|---|---|---|
| *Age, average* ± *St Dev, n = 365* | 31.25±9.08 | 30.99 ± 8.82 | 29.55 ± 9.59 | 31.05 ± 9.04 |
| *Gender, n(%), n = 377* | | | | |
| Male | 10(4.0) | 2(2.10) | 4(12.9) | 16(4.20) |
| Female | 239(96) | 95(97.9) | 27(87.1) | 361(95.8) |
| **Current post/responsibility, n (%)** | **n = 231** | **n = 96** | **n = 31** | **n = 358** |
| Doctors (Specialists, Physicians, medical officers) | 18(7.8) | 0(0) | 0(0) | 18(5) |
| Nurses (Staff nurse, nursing in-charges, Matrons) | 156(67.5) | 51(53.1) | 11(35.5) | 218(60.9) |
| Auxiliary nurse midwives | 46(19.9) | 38(39.6) | 10(32.3) | 94(26.3) |
| Paramedics and others (Health assistants, Auxiliary health workers, etc.) | 11(4.8) | 7(7.3) | 10(32.3) | 28(7.8) |
| **Years of working experience,** average ± SD, n = 376 | 9.29±8.26 | 9.16±8.75 | 7.68±9.71 | 9.12±8.50 |
| **Received neonatal resuscitation training before,** n (%), n = 372 | 98(40.2) | 65(67) | 16(51.6) | 179(48.1) |
| **Resuscitation experience in the past,** n = 375 | 135(54.7) | 65(67) | 22(71) | 222(59.2) |
| **Attended deliveries in the past, n = 365** | 90(38) | 49(50.5) | 20(64.5) | 159(43.6) |

## Discussion

This is the first scaled-up pre-post study to assess the uptake and retention of knowledge and skills on neonatal resuscitation in Nepal. We found an improvement in knowledge and skills of neonatal resuscitation after HBB training together with QI intervention. The knowledge and skills deteriorated over time but were not below the set standard even after six months. The deterioration of knowledge and skills was not associated with any of the learners' characteristics. Almost 51.4% of the health workers in our study were not trained on neonatal resuscitation at the time of initiation of intervention. The proportion of untrained health workers is, however lower than that reported by a study in India and Kenya where 76% of the health workers from 71 health facilities were found untrained [2]. Even though the proportion of health workers untrained in neonatal resuscitation is lower compared to India and Kenya, this is still high and warrants rapid scale-up of neonatal resuscitation training in Nepal.

The finding of our study that knowledge and skills can be improved through the HBB training is similar to the findings of other studies conducted in similar settings [2, 20, 25, 26]. A pre-post study in multiple sites in India and Kenya reported improved neonatal knowledge and skills after HBB training [2]. Carlo WA et al. reported that knowledge scores improved from 57% to 80% and skill scores from 43% to 88% after training in Zambia [26]. Some of the studies have reported improved and even retained knowledge and skills over time [15, 17]. KC et al. reported that the knowledge was improved during basic training and was retained 6 months after the training [15].

We found the deterioration of knowledge and skill with time, similar to the findings with some studies [2, 14, 22, 26, 27]. The proportion of birth attendants passing the OSCE B skill test decreased from post-initial training (99%) to pre-refresher training (81%) in India and Kenya [2]. Carlo WA reported a decrease in knowledge and skill scores after six months of HBB training in Zambia [26]. Even pediatric residents in New York University Langone Medical Center showed skill decay 7 and 10 months after the initial neonatal resuscitation program (NRP) training [27]. The systematic review of newborn resuscitation training revealed that 5 out 10 (50%) studies reported a decrease in knowledge and skills within the period of 1 month to 2 years after basic training [28]. Although the deterioration was observed as a whole, the

**Table 2. Average knowledge and skill test scores by volume of the hospital and individual hospital.**

| Hospitals | Basic training, N = 380 | | Difference, (p-value) | Pre-refresher test, Average ± SD | Difference (p-value) | |
|---|---|---|---|---|---|---|
| | Pre-test score, average ± SD | Post-test score, average ± SD | | | Post-basic vs pre-refresher | Pre-basic vs pre-refresher |
| **A. Knowledge test scores** | | | | | | |
| **A1. By volume of hospital** | | | | | | |
| High volume (n = 252) | 14.10 ± 2.65 | 15.87 ± 2.21 | 0.000 | 15.50 ± 1.97 | 0.001 | 0.000 |
| Medium volume (n = 97) | 14.25 ± 2.52 | 15.93 ± 1.90 | 0.000 | 14.93 ± 2.49 | 0.000 | 0.001 |
| Low volume (n = 31) | 13.90 ± 3.49 | 16.16 ± 1.39 | 0.000 | 15.12 ± 2.26 | 0.029 | 0.010 |
| *Total (n = 380)* | *14.12 ± 2.69* | *15.91 ± 2.08* | *0.000* | *15.33 ± 2.15* | *0.000* | *0.000* |
| **A2. By individual hospital** | | | | | | |
| Hospital1 (n = 15) | 14.80 ± 2.07 | 15.07 ± 3.63 | 0.105 | 15.33 ± 2.12 | 0.454 | 0.106 |
| Hospital2 (n = 53) | 15.30 ± 1.87 | 16.21 ± 1.45 | 0.000 | 16.09 ± 0.52 | 0.381 | 0.001 |
| Hospital3 (n = 11) | 14.91 ± 1.70 | 16.27 ± 1.27 | 0.007 | 15.09 ± 2.46 | 0.109 | 0.198 |
| Hospital 4 (n = 71) | 13.37 ± 2.98 | 14.94 ± 2.94 | 0.000 | 15.25 ± 1.99 | 0.354 | 0.000 |
| Hospital 5 (n = 30) | 13.60 ± 2.95 | 16.63 ± 0.76 | 0.000 | 14.76 ± 1.92 | 0.000 | 0.035 |
| Hospital 6 (n = 44) | 13.43 ± 2.51 | 16.14 ± 1.26 | 0.000 | 14.50 ± 2.68 | 0.000 | 0.002 |
| Hospital 7 (n = 84) | 14.32 ± 2.56 | 16.31 ± 2.07 | 0.000 | 15.88 ± 1.53 | 0.000 | 0.000 |
| Hospital 8 (n = 19) | 13.84 ± 2.94 | 16 ± 1.15 | 0.001 | 15.73 ± 1.40 | 0.305 | 0.000 |
| Hospital9 (n = 20) | 13.35 ± 4.10 | 16.10 ± 1.48 | 0.000 | 15.15 ± 2.20 | 0.055 | 0.005 |
| Hospital10 (n = 33) | 14.82 ± 1.87 | 15.64 ± 1.71 | 0.003 | 14.45 ± 3.38 | 0.042 | 0.269 |
| **B. Skill test (OSCE B scores),** | | | | | | |
| **B1. By volume of hospitals** | | | | | | |
| High volume (n = 188) | | 17.06 ± 1.69 | | 16.45 ± 1.97 | 0.000 | |
| Medium volume (n = 86) | | 16.78 ± 2.09 | | 16.56 ± 1.93 | 0.548 | |
| Low-volume (n = 12) | | 17.08 ± 0.90 | | 15.41 ± 2.57 | 0.080 | |
| **Total (n = 286)** | | *16.98 ± 1.79* | | *16.44 ± 1.99* | *0.000* | |
| **B2. By individual hospitals** | | | | | | |
| Hospital 1 (n = 14) | | 17.43 ± 1.60 | | 15.35 ± 1.21 | 0.000 | |
| Hospital 2 (n = 49) | | 17.04 ± 2.24 | | 16.20 ± 1.75 | 0.001 | |
| Hospital 3 (n = 11) | | 17.18 ± 0.87 | | 16.00 ± 1.67 | 0.074 | |
| Hospital 4 (n = 57) | | 17.35 ± 1.39 | | 16.22 ± 2.53 | 0.001 | |
| Hospital 5 (n = 28) | | 17.61 ± 0.73 | | 16.57 ± 2.57 | 0.023 | |
| Hospital 7 (n = 82) | | 16.87 ± 1.47 | | 16.75 ± 1.60 | 0.279 | |
| Hospital 8 (n = 14) | | 14.86 ± 1.95 | | 17.50 ± 1.34 | 0.000 | |
| Hospital 10 (n = 30) | | 16.60 ± 2.60 | | 16.70 ± 1.48 | 0.486 | |

mean score during refresher training (16.44) was higher than the deteriorated skill level of midwives in rural Ghana (14.6) [22]. Similarly, the proportion of participants who passed the OSCE B test pre-refresher in our study (87.8%) is slightly higher than that in India and Kenya (81%) [2]. The deterioration of knowledge and skills was not associated with any of the learner characteristics (Table 4) and skills (Table 5). It indicates that the selection of participants for HBB training is not worth deciding based on the health workers' characteristics in public hospitals.

**Table 3. Training outcomes knowledge and skill test for trainees by volume of hospitals.**

| Hospital level | Passed Basic training | | Difference -pre vs post (p- value) | Passed refresher training (pre-refresher) | Change over time (post-basic vs pre- refresher pre) | Change over time (pre-basic vs pre-refresher) |
|---|---|---|---|---|---|---|
| | Pre-test | Post-test | | | | |
| *A. Knowledge test, n (%)* | | | | | | |
| High-volume (n = 252) | 170 (67.5) | 226 (89.7) | **0.000** | 223(88.5) | 0.380 | **0.000** |
| Medium-volume (n = 97) | 67(69.1) | 91 (93.8) | **0.000** | 81(83.5) | **0.011** | **0.006** |
| Low-volume (n = 31) | 21(67.7) | 29(93.5) | **0.004** | 25(80.6) | 0.109 | 0.145 |
| *Total, n = 380* | *258 (67.9)* | *346 (91.1)* | *0.000* | *329(86.6)* | *0.023* | *0.000* |
| *B. Skill test–OSCE B, n(%)* | | | | | | |
| High-volume (n = 188) | | 180 (95.7) | | 164(87.2) | **0.001** | |
| Medium-volume (n = 86) | | 78(90.7) | | 77(89.5) | 0.500 | |
| Low-volume (n = 12) | | 12(100) | | 10(83.3) | *NE* | |
| *Total, n = 286* | | *270 (94.4)* | | *251(87.8)* | *0.002* | |

We believe that the deterioration of knowledge and skills in our study is not scientifically significant since the overall score level was maintained above the standard at pre-refresher training. Even though the knowledge and skills deteriorated with time, we found that it had been well maintained above the standard in all of the hospitals compared to the baseline performance. The mean knowledge score before refresher training was higher than the baseline performance in high-volume, medium-volume, and low-volume hospitals. Similarly, a higher mean knowledge score was maintained before refresher training in 7 out of 10 hospitals compared to the baseline performance. The finding is consistent with a study conducted in Tanzania where the proportion of trainees passing knowledge tests increased from 18% before HBB training to 74% seven months after HBB training [13]. Similarly, the deteriorated level of skill level in our study was still well above the set standard. This maintenance of the level of knowledge and skill above the standard can be attributed to the QI intervention package.

No association of the deterioration of knowledge and skills with any of the learners' background characteristics in our study might be due to the involvement of all participants in the quality improvement process after training. After the training, all participants were involved in the daily bag and mask skill checks on the mannequin, and PDSA meetings that provided an equal opportunity for retention of knowledge and skills. The HBB is based on the simplified resuscitation steps [29], and we have demonstrated that regular practice can improve knowledge and skills irrespective of the background characteristics of the health workers.

Our study has three considerable strengths. First, it involves a relatively large number of participants compared to most of the previous studies. Second, the training participants were selected by the hospitals themselves who represent the average level of participants in Nepal. Also, the intervention was introduced in the existing set-up of hospitals, without any modifications in structure or management; the study hospitals represent the average hospital level in Nepal. Therefore, the findings can be fairly generalized to similar hospital settings.

**Table 4. Factors associated with deterioration of knowledge (post-basic vs pre-refresher).**

| Characteristics | Attended post basic and pre refresher knowledge test, N = 380 | | Univariate OR(95% CI):p value | Multivariate aOR (95% CI):p value |
|---|---|---|---|---|
| | Deteriorated or unimproved (n = 51) | Retained or improved (n = 329) | | |
| **1. Level of associated hospitals, n = 380** | | | | |
| High volume (n = 252) | 29 (11.5) | 223(88.5) | Ref | Ref |
| Medium-volume (n = 97) | 16 (16.5) | 81(83.5) | 1.519 (0.784.2942): 0.215 | 1.769 (0.810–3.866): 0.153 |
| Low-volume (n = 31) | 6 (19.4) | 25(80.6) | 1.846 (0.699–4.875): 0.216 | 1.544(0.464–5.132): 0.479 |
| **2. Current position, N = 358** | | | | |
| Doctors, n = 18 | 2 (11.9) | 16 (88.9) | Ref | Ref |
| Nurses, n = 218 | 21 (9.6) | 197(90.4) | 0.853 (0.183–3.967): 0.839 | 0.625 (0.126–3.094): 0.564 |
| ANMs, n = 94 | 14 (14.9) | 80(85.1) | 1.400 (0.290–6.769): 0.676 | 1.003 (0.186–5.408): 0.998 |
| Others, n = 28 | 11 (39.3) | 17(60.7) | 5.176 (0.990–27.064): 0.051 | 2.278 (0.331–15.662): 0.403 |
| **3. Age, n (%)** | 51(13.4) | 329(86.6) | 1.014 (0.982–1.047): 0.400 | 1.000 (0.927–1.079): 0.997 |
| **4. Gender, n(%), n = 377** | | | | |
| Male, n = 16 | 6(37.5) | 10(62.5) | 4.323 (1.497–12.478): **0.007** | 1.178 (0.260–5.343): 0.832 |
| Female, n = 361 | 44(12.2) | 317(87.8) | Ref | Ref |
| **5. Years of working experience, n = 376** | 51(13.4) | 329(86.6) | 1.016 (0.983–1.051): 0.343 | 1.031 (0.954–1.115): 0.442 |
| **6. Received previous HBB training n(%), n = 372** | | | | |
| Yes, n = 179 | 21(11.7) | 158(88.3) | Ref | Ref |
| No, n = 193 | 28(14.5) | 165(85.5) | 1.277 (0.696–2.341): 0.430 | 1.150 (0.540–2.445): 0.717 |
| **7. Resuscitated Newborn with bag and mask before, n(%) n = 375** | | | | |
| Yes, n = 222 | 23(10.4) | 199(89.6) | Ref | |
| No, n = 153 | 27(17.6) | 126(82.4) | 1.854 (1.018–3.376): **0.043** | 1.803 (0.810–4.017): 0.149 |
| **8. Attended Deliveries in the past, n(%) n = 375** | | | | |
| Yes, n = 159 | 16(10.1) | 143(89.9) | Ref | |
| No, n = 206 | 33(16.0) | 173(84.0) | 1.705 (0.902–3.223): 0.101 | 1.414 (0.629–3.179): 0.402 |

This study has some limitations. Firstly, the intention to perform the paired comparison of training participants led to a reduced number of participants for analysis. Out of 798 participants enrolled in the basic training, only 380 (47.6%) were included in the analysis. Second, there is a possibility of the Hawthorne effect since the participants in the basic training were informed about the refresher training planned six months later. Thirdly, the duration gap between basic and refresher training was not the same for all of the hospitals. Although the refresher training was planned to be conducted exactly after six months of basic training, it could not happen in some of the hospitals. It was because the training dates were decided by the respective hospitals, and the study team had not much control over it. Fourth, we did not study the association of the specific QI tools in the package with uptake and retention of knowledge and skills. More insight on ways to improve QI interventions itself could have been generated concerning uptake and retention of neonatal resuscitation knowledge and skills. Further studies can generate evidence on the roles of specific QI tools on uptake and retention of knowledge and skills.

Nepal is committed to reducing neonatal mortality from the current rate of 21/1000 live births to 11/1000 live births by 2035 [23]. Intrapartum hypoxic events being a major cause of neonatal mortality in the country, neonatal resuscitation training should be scaled up to upgrade health workers' competency on neonatal resuscitation. The HBB training should be accompanied by QI interventions for improved retention of knowledge and skills over time.

**Table 5. Factors associated with deterioration of skills (post-basic vs pre-refresher).**

| Characteristics | Attended post basic and pre refresher skill test, N = 286 | | Univariate OR(95% CI): p value | Multivariate aOR (95% CI): p value |
|---|---|---|---|---|
| | Deteriorated or unimproved (n = 35) | Retained or improved (n = 251) | | |
| **1. Level of associated hospitals, n = 286** | | | | |
| High volume (n = 188) | 24(12.8) | 164(87.2) | Ref | Ref |
| Medium-volume (n = 86) | 9(10.5) | 77(89.5) | 0.799(0.354–1.800): 0.588 | 0.886(0.334–2.353): 0.808 |
| Low-volume (n = 12) | 2(16.7) | 10(83.3) | 1.367(0.282–6.618): 0.698 | 1.959(0.313–12.282): 0.473 |
| **2. Current position, N = 272** | | | | |
| Doctors (n = 15) | 2(13.3) | 13(86.7) | Ref | |
| Nurses (n = 182) | 20(11.0) | 162(89.0) | 0.802(0.169–3.817): 0.782 | 1.470(0.171–12.660): 0.726 |
| ANMs (n = 64) | 8(12.5) | 56(87.5) | 0.929(0.176–4.897): 0.930 | 1.893(0.189–18.917): 0.587 |
| Others (n = 11) | 2(18.2) | 9(81.8) | 1.444(0.171–12.232): 0.736 | 3.952(0.224–69.852): 0.348 |
| **3. Age, n = 275** | | | 1.016(0.977–1.056): 0.437 | 1.058(0.940–1.191): 0.350 |
| **4. Gender, n(%), n = 286** | | | | |
| Male, n = 5 | 0(0) | 5(100) | *NE* | NE |
| Female, n = 281 | 35(12.5) | 246(87.5) | *Ref* | Ref |
| **5. Years of working experience, n = 286** | | | 1.011 (6.971–1.054): 0.587 | 0.910(0.790–1.049): 0.193 |
| **6. Received previous HBB training n(%), n = 283** | | | | |
| Yes, n = 142 | 16(11.3) | 126(88.7) | Ref | Ref |
| No, n = 141 | 19(13.5) | 122(86.5) | 1.226(0.603–2.495): 0.573 | 1.175(0.459–3.008): 0.736 |
| **7. Resuscitated Newborn with Bag and Mask before, n(%) n = 284** | | | | |
| Yes, n = 168 | 22(13.1) | 146(86.9) | Ref | Ref |
| No, n = 116 | 13(11.2) | 103(88.8) | 0.838(0.403–1.739): 0.634 | 0.976(0.333–2.857): 0.964 |
| **8. Attended Deliveries in the past, n(%) n = 276** | | | | |
| Yes, n = 114 | 18(15.8) | 96(84.2) | Ref | Ref |
| No, n = 162 | 13(8.0) | 149(92.0) | **0.465(0.218–0.993): 0.048** | 0.501(0.197–1.310): 0.161 |

## Conclusion

HBB training together with QI tools improve health workers' knowledge and skills on neonatal resuscitation, irrespective of the size and type of hospitals. The knowledge and skills deteriorate over time but do not fall below the standard. The level of knowledge and skills can be well maintained above the standard compared to baseline status (pre-basic training); indicating their capacity to retain the level of competency required to perform neonatal resuscitation whenever required. The deterioration of knowledge and skills are not associated with the learner's background characteristics and hence should not be considered as selection criteria for training. The HBB training together with QI interventions can be scaled up in other public hospitals of Nepal and other similar settings. Further studies are indicated to identify more pragmatic ways for improved retention of knowledge and skills on neonatal resuscitation.

## Supporting information

**S1 Text. Quality improvement intervention guideline.**
(PDF)

**S2 Text. NePeriQIP training registration and course evaluation form.**
(PDF)

**S1 Dataset.**
(CSV)

## Acknowledgments

We are grateful to Anna Bergstrom and Leif Erikson for facilitating the master training of trainers. We would like to thank Abhishek Gurung, Deepak Jha, Elisha Joshi, Gambhir Shrestha, Sunil Gajurel, Alyza Dhanwantary for facilitating training in hospitals. We thank the Department of Health Services, Child Health Division for administrative and technical support during training. We acknowledge the active participation of health service providers from participating hospitals during training.

## Author Contributions

**Conceptualization:** Ashish K. C., Mats Malqvist.

**Data curation:** Omkar Basnet.

**Formal analysis:** Dipak Raj Chaulagain, Mats Malqvist.

**Funding acquisition:** Ashish K. C., Mats Malqvist.

**Investigation:** Dipak Raj Chaulagain, Ashish K. C., Johan Wrammert.

**Methodology:** Dipak Raj Chaulagain, Ashish K. C., Johan Wrammert, Olivia Brunell, Mats Malqvist.

**Project administration:** Dipak Raj Chaulagain, Olivia Brunell.

**Software:** Ashish K. C., Omkar Basnet.

**Supervision:** Dipak Raj Chaulagain, Johan Wrammert, Olivia Brunell, Mats Malqvist.

**Visualization:** Johan Wrammert, Mats Malqvist.

**Writing – original draft:** Dipak Raj Chaulagain.

**Writing – review & editing:** Ashish K. C., Johan Wrammert, Olivia Brunell, Omkar Basnet, Mats Malqvist.

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
