## [Decision Letter · Decision Letter 0]

16 Nov 2020

PONE-D-20-15505

Effect of a scaled-up quality improvement intervention on service providers' competence on neonatal resuscitation in simulated settings in public hospitals: a pre-post study in Nepal.

PLOS ONE

Dear Dr. Chaulagain,

Thank you for submitting your manuscript to PLOS ONE. After careful consideration, we feel that it has merit but does not fully meet PLOS ONE’s publication criteria as it currently stands. Therefore, we invite you to submit a revised version of the manuscript that addresses the points raised during the review process.

We look forward to receiving your revised manuscript.

Kind regards,

Anne Lee Solevåg, M.D., Ph.D.

Academic Editor

PLOS ONE

Journal Requirements:

2. Please include additional information regarding the survey or questionnaire used in the study and ensure that you have provided sufficient details that others could replicate the analyses. For instance, if you developed a questionnaire as part of this study and it is not under a copyright more restrictive than CC-BY, please include a copy, in both the original language and English, as Supporting Information.  If the original language is written in non-Latin characters, for example Amharic, Chinese, or Korean, please use a file format that ensures these characters are visible.

4.In your Data Availability statement, you have not specified where the minimal data set underlying the results described in your manuscript can be found. PLOS defines a study's minimal data set as the underlying data used to reach the conclusions drawn in the manuscript and any additional data required to replicate the reported study findings in their entirety. All PLOS journals require that the minimal data set be made fully available. For more information about our data policy, please see http://journals.plos.org/plosone/s/data-availability.

6.Thank you for stating the following in the Financial Disclosure section:

[Funding was obtained from the Vetenskapsrådet (SE), the Laerdal Foundation for

Acute Medicine, Norway, and Einhorn Family Foundation, Sweden. AKC was the grantee of

Vetenskapsrådet (SE).MM was the grantee of Laerdal Foundation for Acute Medicine, Norway,

and Einhorn Family Foundation, Sweden. The funders had no role in study design, data collection

and analysis, decision to publish, or preparation of the manuscript.].   

We note that one or more of the authors are employed by a commercial company: Golden Community, Lalitpur, Nepal

7. Please note that in order to use the direct billing option the corresponding author must be affiliated with the chosen institute. Please either amend your manuscript to change the affiliation or corresponding author, or email us at plosone@plos.org with a request to remove this option.

Reviewers' comments:

Reviewer's Responses to Questions

**Comments to the Author**

1. Is the manuscript technically sound, and do the data support the conclusions?

Reviewer #1: Yes

Reviewer #2: Yes

2. Has the statistical analysis been performed appropriately and rigorously? 

Reviewer #1: Yes

Reviewer #2: Yes

3. Have the authors made all data underlying the findings in their manuscript fully available?

Reviewer #1: Yes

Reviewer #2: Yes

4. Is the manuscript presented in an intelligible fashion and written in standard English?

Reviewer #1: No

Reviewer #2: Yes

5. Review Comments to the Author

Reviewer #1: Abstract: the authors summarize the main research question and key findings succinctly.

Introduction: the authors do a sufficiently good job in setting the stage for the paper through several references quoted, leading finally to the research question. I would like to advice the authors to increase references for QI approaches and outcomes. The text in rows 87-89 requires revision. My suggestion would be to remove QI from the training approach and to note it as an implementation approach.

Intervention: Please clarify if the external mentors were employed by the public health sector or from the private sector

Results: Knowledge and skills scores were analyzed by hospital size, individual hospital, but the results were used to interpret the findings in a limited fashion. An interesting finding from this analysis noted skills deterioration was higher for high volume hospitals. This finding would benefit from understanding why that was so.

Discussion: I think the authors have done a really good job in setting up interventions and have a very interesting and valuable results from the data analyzed. Some of the results could be worth discussing. The ones that I would recommend for the authors to add their observations on the lack of statistical difference in retention of skills over time in high, medium and low volume hospitals and among the different providers under current position. Traditionally, we would have expected more opportunities for practice / implementation would have meant more retention of skills. That is not demonstrated by this study.

Overall write-up is clear and appealing and of value. There are a couple of minor typos that could benefit from a thorough copy-editing by the authors

Reviewer #2: This is an interesting study evaluating the pre-post study design in 12 public hospitals of Nepal that is assessing the knowledge and skills of trainees on neonatal resuscitation. There is a clear need for increased knowledge and skills of neonatal resuscitation to help improve outcomes of neonates. This study evaluated a year+ long effort to train many providers on HBB and test their pre-post knowledge. It showed that the HBB with QI package is helpful in this setting to improve knowledge retention and hope future efforts will continue to build capacity there for this vulnerable population.

6. PLOS authors have the option to publish the peer review history of their article (what does this mean?). If published, this will include your full peer review and any attached files.

Reviewer #1: **Yes: **Neena Khadka

Reviewer #2: No

---

## [Author Response · Author response to Decision Letter 0]

27 Jan 2021

Response to reviewer has been uploaded as a separate file

---

## [Editor Report · Decision Letter 1]

5 Feb 2021

PONE-D-20-15505R1

Effect of a scaled-up quality improvement intervention on health workers' competence on neonatal resuscitation in simulated settings in public hospitals: a pre-post study in Nepal.

PLOS ONE

Dear Dipak Raj Chaulagain,

Thank you for submitting your manuscript to PLOS ONE. After careful consideration, we feel that it does meet PLOS ONE’s publication criteria. However, we would like your opinion about two remaining questions that the academic editor has. Therefore, we invite you to submit a revised version of the manuscript that addresses the points raised during the review process.

ACADEMIC EDITOR: 

1. In the ethics statement, if verbal consent was obtained, how it was documented and witnessed

2. Reviewer comment: -I would like to advice the authors to increase references for QI approaches and outcomes

A rebuttal letter that responds to each point raised by the academic editor. You should upload this letter as a separate file labeled 'Response to Reviewers'.A marked-up copy of your manuscript that highlights changes made to the original version. You should upload this as a separate file labeled 'Revised Manuscript with Track Changes'.An unmarked version of your revised paper without tracked changes. You should upload this as a separate file labeled 'Manuscript'.

We look forward to receiving your revised manuscript.

Kind regards,

Anne Lee Solevåg, M.D., Ph.D.

Academic Editor

PLOS ONE

Additional Editor Comments (if provided):

Dear Dipak Raj Chaulagain,

Thank you for submitting your revisions and answers to the reviewer and editorial office's queries and comments.

After having read the revised manuscript, I find that a couple of issues have been incompletely addressed in the revision:

1. In the ethics statement, if verbal consent was obtained, how it was documented and witnessed

2. Reviewer comment: -I would like to advice the authors to increase references for QI approaches and outcomes.

Regards,

Anne Lee Solevåg, M.D., Ph.D.

PLOS ONE

Reviewers' comments:

No external review of the revised manuscript

---

## [Author Response · Author response to Decision Letter 1]

1 Apr 2021

Anne Lee Solevåg, 01 April 2021

Academic Editor

PLOS ONE

Re: Comments addressed for the manuscript "Effect of a scaled-up quality improvement intervention on health workers' competence on neonatal resuscitation in simulated settings in public hospitals: a pre-post study in Nepal" (PONE-D-20-15505R1)

Dear Anne Lee,

Thank you very much for providing opportunity to revise our manuscript for the second time. We have attempted to address all the comments in our revised manuscript. Please find the responses (in bold) to each comment below;

Academic editor comments

1. In the ethics statement, if verbal consent was obtained, how it was documented and witnessed

Response - Additional details regarding participant consent have been specified in the ethics statement section of the manuscript (Page11, line 235-243). 

All participants were provided with verbal information on their participation in the study. The training registration and evaluation form (S2 Text), which was used to evaluate trainees' knowledge and skills in this study, was used for the documentation of their consent. The participants filled and signed the 'training registration and evaluation form ' after agreeing to participate in the study. The process of obtaining consent was completed in the presence of hospital management in the respective hospitals. The signed individual training registration and evaluation forms have been securely stored at the central research office in Kathmandu. 

2. Reviewer comment: -I would like to advice the authors to increase references for QI approaches and outcomes

The references for QI approaches and outcomes have been increased (page 4,5, line 85-91)

Yours Sincerely,

Dipak Raj Chaulagain, MPH (Corresponding author)

Department of Women's and Children's Health, Uppsala University, Uppsala, Sweden

Email: dipak.chaulagain@kbh.uu.se

---

## [Editor Report · Decision Letter 2]

14 Apr 2021

Effect of a scaled-up quality improvement intervention on health workers' competence on neonatal resuscitation in simulated settings in public hospitals: a pre-post study in Nepal.

PONE-D-20-15505R2

Dear Dr. Dipak Raj Chaulagain,

We’re pleased to inform you that your manuscript has been judged scientifically suitable for publication and will be formally accepted for publication once it meets all outstanding technical requirements.

Kind regards,

Anne Lee Solevåg, M.D., Ph.D.

Academic Editor

PLOS ONE
---

## [Editor Report · Acceptance letter]

20 Apr 2021

PONE-D-20-15505R2 

Effect of a scaled-up quality improvement intervention on health workers' competence on neonatal resuscitation in simulated settings in public hospitals: a pre-post study in Nepal.  

Dear Dr. Chaulagain:

I'm pleased to inform you that your manuscript has been deemed suitable for publication in PLOS ONE. Congratulations! Your manuscript is now with our production department. 

Kind regards, 

on behalf of

Dr. Anne Lee Solevåg 

Academic Editor

PLOS ONE